# Quantum theory based on real numbers can be experimentally falsified

Marc-Olivier Renou[1], David Trillo[2], Mirjam Weilenmann[2], Thinh P. Le[2], Armin Tavakoli[2,3], Nicolas Gisin[4,5], Antonio Acín[1,6] & Miguel Navascués[2 ✉]

Although complex numbers are essential in mathematics, they are not needed to describe physical experiments, as those are expressed in terms of probabilities, hence real numbers. Physics, however, aims to explain, rather than describe, experiments through theories. Although most theories of physics are based on real numbers, quantum theory was the first to be formulated in terms of operators acting on complex Hilbert spaces[1,2]. This has puzzled countless physicists, including the fathers of the theory, for whom a real version of quantum theory, in terms of real operators, seemed much more natural[3]. In fact, previous studies have shown that such a 'real quantum theory' can reproduce the outcomes of any multipartite experiment, as long as the parts share arbitrary real quantum states[4]. Here we investigate whether complex numbers are actually needed in the quantum formalism. We show this to be case by proving that real and complex Hilbert-space formulations of quantum theory make different predictions in network scenarios comprising independent states and measurements. This allows us to devise a Bell-like experiment, the successful realization of which would disprove real quantum theory, in the same way as standard Bell experiments disproved local physics.

Without qualification, the question of whether complex numbers are necessary for natural sciences, and, more concretely, for physics, must be answered in the negative: physics experiments are described by the statistics they generate, that is, by probabilities, and hence real numbers, thus, there is no need for complex numbers. The question becomes meaningful, however, when considering a specific theoretical framework, designed to explain existing experiments and make predictions about future ones. Whether complex numbers are needed within a theory to correctly explain experiments, or whether real numbers only are sufficient, is not straightforward. Complex numbers are sometimes introduced in electromagnetism to simplify calculations: one might, for instance, regard the electric and magnetic fields as complex vector fields to describe electromagnetic waves. However, this is just a computational trick. We wonder whether the same can be said about complex numbers in quantum theory.

In its Hilbert space formulation, quantum theory is defined in terms of the following postulates[5,6]. (1) For every physical system $S$, there corresponds a Hilbert space $\mathcal{H}_S$ and its state is represented by a normalized vector $\phi$ in $\mathcal{H}_S$, that is, $\langle\phi|\phi\rangle = 1$. (2) A measurement $\Pi$ in $S$ corresponds to an ensemble $\{\Pi_r\}_r$ of projection operators, indexed by the measurement result $r$ and acting on $\mathcal{H}_S$, with $\sum_r \Pi_r = \mathbb{I}_S$. (3) Born rule: if we measure $\Pi$ when system $S$ is in state $\phi$, the probability of obtaining result $r$ is given by $\Pr(r) = \langle\phi|\Pi_r|\phi\rangle$ (4) The Hilbert space $\mathcal{H}_{ST}$ corresponding to the composition of two systems $S$ and $T$ is $\mathcal{H}_S \otimes \mathcal{H}_T$. The operators used to describe measurements or transformations in system $S$ act trivially on $\mathcal{H}_T$ and vice versa. Similarly, the state representing two

independent preparations of the two systems is the tensor product of the two preparations.

This last postulate has a key role in our discussions: we remark that it even holds beyond quantum theory, specifically for space-like separated systems in some axiomatizations of quantum field theory[7–10] (Supplementary Information).

As originally introduced by Dirac and von Neumann[1,2], the Hilbert spaces $\mathcal{H}_S$ in postulate (1) are traditionally taken to be complex. We call the resulting postulate ($1_c$). The theory specified by postulates ($1_c$) and (2)–(4) is the standard formulation of quantum theory in terms of complex Hilbert spaces and tensor products. For brevity, we will refer to it simply as 'complex quantum theory'. Contrary to classical physics, complex numbers (in particular, complex Hilbert spaces) are thus an essential element of the very definition of complex quantum theory.

Owing to the controversy surrounding their irruption in mathematics and their almost total absence in classical physics, the occurrence of complex numbers in quantum theory worried some of its founders, for whom a formulation in terms of real operators seemed much more natural ("What is unpleasant here, and indeed directly to be objected to, is the use of complex numbers. $\Psi$ is surely fundamentally a real function." (Letter from Schrödinger to Lorentz, 6 June 1926; ref. [3])). This is precisely the question we address in this work: whether complex numbers can be replaced by real numbers in the Hilbert space formulation of quantum theory without limiting its predictions. The resulting 'real quantum theory', which has appeared in the literature under various names[11,12], obeys the same postulates (2)–(4) but assumes real

[1]ICFO-Institut de Ciencies Fotoniques, The Barcelona Institute of Science and Technology, Castelldefels (Barcelona), Spain. [2]Institute for Quantum Optics and Quantum Information (IQOQI) Vienna, Austrian Academy of Sciences, Vienna, Austria. [3]Institute for Atomic and Subatomic Physics, Vienna University of Technology, Vienna, Austria. [4]Group of Applied Physics, University of Geneva, Geneva, Switzerland. [5]Schaffhausen Institute of Technology–SIT, Geneva, Switzerland. [6]ICREA-Institució Catalana de Recerca i Estudis Avançats, Barcelona, Spain. ✉e-mail: miguel.navascues@oeaw.ac.at

Hilbert spaces $\mathcal{H}_S$ in postulate (1), a modified postulate that we denote by $(1_\mathbb{R})$.

If real quantum theory led to the same predictions as complex quantum theory, then complex numbers would just be, as in classical physics, a convenient tool to simplify computations but not an essential part of the theory. However, we show that this is not the case: the measurement statistics generated in certain finite-dimensional quantum experiments involving causally independent measurements and state preparations do not admit a real quantum representation, even if we allow the corresponding real Hilbert spaces to be infinite dimensional.

Our main result applies to the standard Hilbert space formulation of quantum theory, through axioms (1)–(4). It is noted, though, that there are alternative formulations able to recover the predictions of complex quantum theory, for example, in terms of path integrals[13], ordinary probabilities[14], Wigner functions[15] or Bohmian mechanics[16]. For some formulations, for example, refs. [17,18], real vectors and real operators play the role of physical states and physical measurements respectively, but the Hilbert space of a composed system is not a tensor product. Although we briefly discuss some of these formulations in Supplementary Information, we do not consider them here because they all violate at least one of the postulates $(1_\mathbb{R})$ and (2)–(4). Our results imply that this violation is in fact necessary for any such model.

It is instructive to address our main question as a game between two players—the 'real' quantum physicist Regina and the 'complex' quantum physicist Conan. Regina is convinced that our world is governed by real quantum theory, whereas Conan believes that only complex quantum theory can describe it. Through a well chosen quantum experiment, Conan aims to prove Regina wrong; that is, to falsify real quantum theory by exhibiting an experiment that this theory cannot explain.

At first, Conan thinks of conducting simple experiments involving a single quantum system. Unfortunately, for any such quantum experiment, Regina can find a real quantum explanation. For instance, if $\rho$ is the complex density matrix that Conan uses to model his experiment, Regina could propose the state

$$
\begin{aligned}
\tilde{\rho} &= \mathrm{Re}(\rho) \otimes \frac{\mathbb{I}}{2} + \mathrm{Im}(\rho) \otimes \frac{1}{2}\begin{pmatrix} 0 & 1 \\ -1 & 0 \end{pmatrix} \\
&= \frac{1}{2}(\rho \otimes |+i\rangle\langle+i| + \rho^* \otimes |-i\rangle\langle-i|),
\end{aligned}
\tag{1}
$$

where $|\pm i\rangle = \frac{1}{\sqrt{2}}(|0\rangle \pm i|1\rangle)$, and the asterisk denotes complex conjugation. The operator $\tilde{\rho}$ is real and positive semidefinite: it is thus a real quantum state. Fig. 1 (left) explains how Regina can analogously define real measurement operators that, acting on $\tilde{\rho}$, reproduce the statistics of any (complex) measurement conducted by Conan on $\rho$. This construction is just one of the infinitely many ways that Regina has to explain the measurement statistics of any single-particle experiment using real operators, but it already implies that real quantum theory cannot be falsified in this scenario. It does not imply, however, that states in real quantum theory are restricted to have this form: they remain arbitrary, as in complex quantum theory.

It is noted that, assuming a fixed Hilbert space dimension, Conan could come up with single-site experiments where real and complex quantum theory differ, for instance, because the former does not satisfy local tomography, or even leads to different experimental predictions (see, for example, ref. [19]). However, as dimension cannot be upper bounded experimentally[20], Regina would be right not to interpret any such experiment as a disproof of real quantum theory. In practice, any experimental system has infinite degrees of freedom: a finite dimension may just be an approximation made to simplify its theoretical description. Hence, to defeat Regina, Conan has to design an experiment in which no explanation using real Hilbert spaces is valid, no matter their dimension.

Conan may next consider experiments involving several distant labs, where phenomena such as entanglement[21] and Bell non-locality[22] can

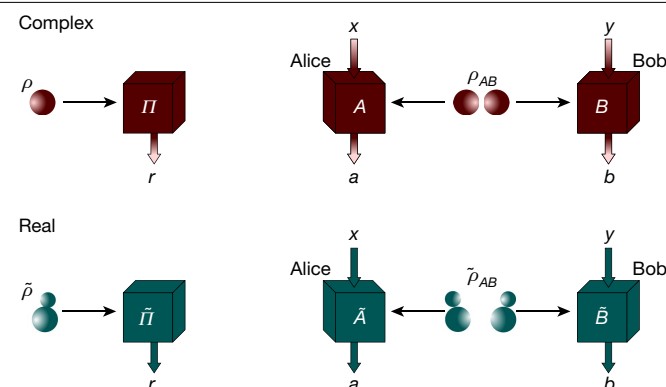

**Fig. 1 | Simulating single-site and multipartite quantum experiments through real quantum theory.** Left: a single-site quantum experiment. A complex quantum system in state $\rho$ is probed via the measurement $\{\Pi_r\}_r$. One way to reproduce the measurement statistics of this experiment using real quantum theory requires adding an extra real qubit: the state $\rho$ is then replaced by the real state $\tilde{\rho}$ in equation (1), while every measurement operator is replaced by the real measurement operator $\widetilde{\Pi}_r = \Pi_r \otimes |i\rangle\langle i| + \Pi_r^* \otimes |-i\rangle\langle-i|$. Using the fact that probabilities are real, and thus $P(r) = P(r)^* = \mathrm{tr}(\rho^* \Pi_r^*)$, it is straightforward that $P(r) = \mathrm{tr}(\rho \Pi_r) = \mathrm{tr}(\tilde{\rho}\widetilde{\Pi}_r)$. It is noted that this construction doubles the Hilbert space dimension of the original complex quantum system (when finite). This is not surprising, as a complex number is defined by two real numbers, and may just be seen as yet another example of how complex numbers simplify the calculation of experimental predictions, as in electromagnetism. Right: a multipartite quantum experiment. A complex Bell scenario consists of two particles (or systems) distributed between Alice and Bob, who perform local measurements, labelled by $x$ and $y$, and get results $a$ and $b$. By postulates $(1_\mathbb{R})$–(4), a complex Hilbert space is assigned to each particle, and the Hilbert space describing the overall bipartite system is defined by the tensor product of these. The state of the two particles is thus described by an operator $\rho_{AB}$ acting on the joint space, whereas operators $A_{a|x}$ and $B_{b|y}$ acting on each local Hilbert space describe the local measurements. The observed measurement statistics or correlations are described by the conditional probability distribution $P(ab|xy) = \mathrm{tr}(\rho_{AB} A_{a|x} \otimes B_{b|y})$. One way to reproduce these statistics using real quantum theory consists of assigning an extra real qubit to each particle. The quantum state is replaced by the real state $\widetilde{\rho}_{AA'BB'} = \frac{1}{2}(\rho_{AB} \otimes |+i,+i\rangle\langle+i,+i|_{A'B'} + \rho_{AB}^* \otimes |-i,-i\rangle\langle-i,-i|_{A'B'})$, and the local measurements are replaced by the same transformation as before for a single system. The observed statistics are again recovered, that is, $P(ab|xy) = \mathrm{tr}(\widetilde{\rho}_{AB}\widetilde{A}_{a|x} \otimes \widetilde{B}_{b|y})$.

manifest. For simplicity, we focus on the case of two separate labs. A source emits two particles (for example, a crystal pumped by a laser emitting two photons) in a state $\rho_{AB}$, each being measured by different observers, called Alice (A) and Bob (B) (Fig. 1, right). Alice (Bob) conducts measurement $x$ ($y$) on her (his) particle, obtaining the outcome $a$ ($b$). As pointed out by Bell[22], there exist quantum experiments where the observed correlations, encapsulated by the measured probabilities $P(a,b|x,y)$, are such that they cannot be reproduced by any local deterministic model. An experimental realization of such correlations disproves the universal validity of local classical physics.

Next we consider whether Conan could similarly refute real quantum theory via a (complex) quantum Bell experiment. Such an experiment should necessarily violate some Bell inequality; otherwise, one could reproduce the measured probabilities with diagonal (and hence real) density matrices and measurement operators. The mere observation of a Bell violation is, however, insufficient to disprove real quantum theory, as already exemplified by the famous Clauser–Horne–Shimony–Holt (CHSH) Bell inequality[23] $\mathrm{CHSH}(x_1, x_2; y_1, y_2) := \langle A_{x_1} B_{y_1}\rangle + \langle A_{x_1} B_{y_2}\rangle + \langle A_{x_2} B_{y_1}\rangle - \langle A_{x_2} B_{y_2}\rangle \leq 2$. The inequality is derived for a Bell experiment where Alice and Bob perform two measurements with outcomes ±1, and where $A_x$ ($B_y$) denotes the results by Alice (Bob) when performing measurement $x$ ($y$). The maximal quantum violation of this inequality is

$\beta_{\text{CHSH}} = 2\sqrt{2}$ and Alice and Bob can attain it using real measurements on a real two-qubit state.

To find a gap between the predictions of real and complex quantum theory, Conan shall explore more complicated Bell inequalities. A priori, promising candidates are the elegant inequality of ref. [24] or the combination of three CHSH inequalities introduced in refs. [25,26]

$$\text{CHSH}_3 := \text{CHSH}(1, 2\,;\,1, 2) + \text{CHSH}(1, 3\,;\,3, 4) + \text{CHSH}(2, 3\,;\,5, 6) \le 6, \quad (2)$$

designed for a scenario in which Alice and Bob perform three and six measurements, respectively. The maximal violation of inequality (2) is $3\beta_{\text{CHSH}} = 6\sqrt{2}$, which can be attained with complex measurements on qubits[26].

However, none of these Bell inequalities will work: as shown in refs. [4,27,28], real quantum Bell experiments can reproduce the statistics of any quantum Bell experiment, even if conducted by more than two separate parties. Indeed, the construction of equation (1) for single complex quantum systems can be adapted to the multipartite case if we allow the source to distribute an extra qubit for each observer (see Fig. 1, right, for details).

To defeat Regina, Conan may also look for inspiration to other no-go theorems in quantum theory, such as the Pusey–Barrett–Rudolph construction[29] involving states prepared in independent labs subject to joint measurements. Unfortunately, Regina is again able to provide an explanation to such scenarios using real quantum theory (Supplementary Information). At this point, Conan might give up and accept that he will never change Regina's mind. He would not be alone. For years, it was generally accepted that real quantum theory was experimentally indistinguishable from complex quantum theory. In other words, in quantum theory, complex numbers would only be convenient, but not necessary, to make sense of quantum experiments. Next we prove this conclusion wrong.

All it takes for Conan to win the discussion is to go beyond the previous constructions and consider experimental scenarios where independent sources prepare and send entangled states to several parties, who in turn conduct independent measurements[30–34]. Such general network scenarios correspond to the future quantum internet, which will connect many quantum computers and guarantee quantum confidentiality over continental distances. Our results demonstrate how these networks, beyond their practical relevance, open radically new perspectives to solve open questions in the foundations of quantum theory when exploiting the causal constraints associated with their geometries.

To disprove real quantum theory, Conan proposes the network corresponding to a standard entanglement-swapping scenario, depicted in Fig. 2, consisting of two independent sources and three observers: Alice, Bob and Charlie. The two sources prepare two maximally entangled states of two qubits, the first one $\bar{\sigma}_{AB_1}$ distributed to Alice and Bob; and the second $\bar{\sigma}_{B_2C}$, to Bob and Charlie. Bob performs a standard Bell-state measurement on the two particles that he receives from the two sources. This measurement has the effect of swapping the entanglement from Alice and Bob and Bob and Charlie to Alice and Charlie: namely, for each of Bob's four possible outcomes, Alice and Charlie share a two-qubit entangled state. Note that the actual state depends on Bob's outcome, but not on its degree of entanglement, which is always maximal. Alice and Charlie implement the measurements leading to the maximal violation of the CHSH$_3$ inequality (2). For these measurements, the state shared by Alice and Charlie, conditioned on Bob's result, maximally violates the inequality or a variant thereof produced by simple relabellings of the measurement outcomes.

Regina takes up Conan's challenge and seeks to reproduce the statistics predicted by Conan. As she works under the postulates $(1_{\mathbb{R}})$ and $(2)$–$(4)$, she models the experiment of Fig. 2 as follows: each subsystem is represented by a real Hilbert space $\mathcal{H}_S$ for $S = A, B_1, B_2, C$, the states of the two sources are arbitrary real density matrices acting on $\mathcal{H}_A \otimes \mathcal{H}_{B_1}$ and $\mathcal{H}_{B_2} \otimes \mathcal{H}_C$, respectively, and the arbitrary real measurements act

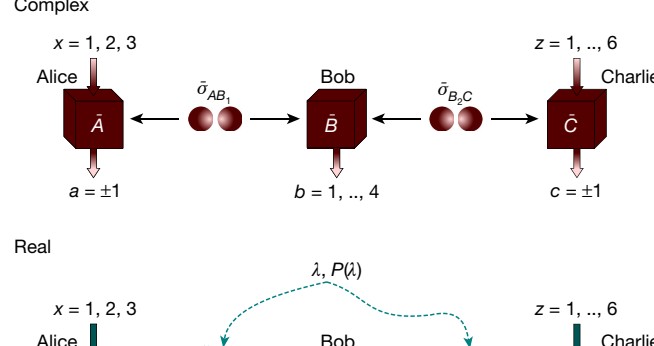

**Fig. 2 | Network scenario separating real and complex quantum theory.** In complex quantum theory (top), two independent sources distribute the two-qubit states $\bar{\sigma}_{AB_1}$ and $\bar{\sigma}_{B_2C}$ to, respectively, Alice and Bob, and Bob and Charlie. At Bob's location, a Bell measurement, of four outputs, is implemented. Alice and Charlie apply the complex measurements leading to the maximal violation of the CHSH$_3$ inequality: three and six measurements with two possible outputs, labelled by ±1. According to quantum physics, the observed correlations read $\bar{P}(abc|xz) = \text{tr}((\bar{\sigma}_{AB_1} \otimes \bar{\sigma}_{B_2C})(\bar{A}_{a|x} \otimes \bar{B}_b \otimes \bar{C}_{c|z}))$. These correlations cannot be reproduced, or even well approximated, when all the states and measurements in the network are constrained to be real operators of arbitrary dimension (bottom). The impossibility still holds if the two preparations are correlated through shared randomness (dashed arrows), resulting in correlations of the form $P(abc|xz) = \sum_\lambda P(\lambda)\,\text{tr}((\tilde{\sigma}^\lambda_{AB_1} \otimes \tilde{\sigma}^\lambda_{B_2C})(\tilde{A}_{a|x} \otimes \tilde{B}_b \otimes \tilde{C}_{c|z}))$, where all operators are real.

on $\mathcal{H}_A$, $\mathcal{H}_{B_1} \otimes \mathcal{H}_{B_2}$ and $\mathcal{H}_C$ respectively. For each choice of states and measurements, she computes the probabilities via the Born rule. Regina's goal is to search over all states and measurements of the aforementioned form, acting on real Hilbert spaces of arbitrary dimension, until she can match Conan's predictions.

However, no construction by Regina is able to reproduce the measurement probabilities $\bar{P}(a, b, c|x, z)$ observed in the experiment. The proof, given in Supplementary Information, exploits the results of ref. [26], where all quantum realizations leading to the maximal quantum value of inequality (2) were characterized. From this characterization, we show that the marginal state shared by Alice and Charlie at the beginning of the experiment cannot be decomposed as a convex combination of real product states[35], as the network depicted in Fig. 2 requires, and thus the statement follows. We moreover show the result to be robust, in the sense that the impossibility of real simulation also holds for non-maximal violations of the inequality (2) between Alice and Charlie. This result settles the argument between Conan and Regina: as she cannot accommodate Conan's experimental observations within the real quantum framework, she must admit her defeat.

A different question now is whether it is experimentally feasible to disprove real quantum theory. To assess this, it is convenient to rephrase our impossibility result in terms of a Bell-type parameter, that is, $\mathcal{T}$, a linear function of the observed correlations. To this aim, we propose the Bell-type functional, defined by the sum of the violations of (the variants of) the CHSH$_3$ inequality for each of Bob's measurement outputs, weighted by the probability of the output. In the ideal entanglement-swapping realization with two-qubit maximally entangled states, the maximal quantum value of CHSH$_3$, equal to $6\sqrt{2}$, is obtained for each of the four outputs by Bob, so $\mathcal{T}$ also attains its maximum quantum value, $\mathcal{T} = 6\sqrt{2} \approx 8.49$. In Supplementary Information, we explain how to reduce the problem of upper bounding $\mathcal{T}$ to a convex optimization problem, making use of the hierarchies[28,36–38],

that we solve numerically[39,40], for real quantum systems, to give $\mathcal{T} \lesssim 7.66$. It remains open whether this upper bound is tight. As the map $\mathcal{T}$ is a linear function of the observed probabilities, the impossibility result holds even when the real simulation is assisted by shared randomness (Fig. 2, bottom). As shown in refs. [41,42], this feature allows one to drop the assumption of independent and identical realizations in multiple-round hypothesis tests.

The setup needed to experimentally falsify real quantum theory is very similar to the bilocality scenario described in ref. [30], for which several experimental implementations have been reported[43–46]. Beating the real bound on $\mathcal{T}$ requires the two distributed states to have each a visibility beyond $\sqrt{7.66/6\sqrt{2}} \approx 0.95$, a value attained in several experimental labs worldwide. The experiment similarly relies on the implementation of a challenging[47] but feasible[48] two-qubit entangled measurement. All things considered, we believe that an experimental disproof of real quantum physics based on the inequality $\mathcal{T}$ is within reach of current quantum technology (see Supplementary Information for more details).

Since the birth of modern science four centuries ago, abstract mathematical entities have played a big role in formalizing physical concepts. Our current understanding of velocity was only possible through the introduction of derivatives. The modern conception of gravity is attributable to the invention of non-Euclidean geometry. Basic notions from representation theory made it possible to formalize the notion of a fundamental particle. Here we considered whether the same holds for the complex numbers. Somewhat surprisingly, we found that there do exist natural scenarios that require the use of complex numbers to account for experimental observations within the standard Hilbert space formulation of quantum theory. As it turns out, some such experiments are within reach of current experimental capabilities, so it is not unreasonable to expect a convincing experimental disproof of real quantum theory in the near future.

From a broader point of view, our results advance the research programme, started in ref. [49], of singling out quantum correlations by demanding maximal performance in a device-independent information-theoretic task. In this regard, our work shows that complex quantum theory outperforms real quantum theory when the non-local game $\mathcal{T}$ is played in the entanglement-swapping scenario. This game can be interpreted as an extension of the adaptive CHSH game proposed in ref. [49], which was recently shown to rule out a number of alternative physical theories in favour of quantum theory[50]. Whether the average score of $\mathcal{T}$ is maximized by complex quantum theory, or whether any physical theory other than complex quantum theory must necessarily produce a lower score are intriguing questions that we leave open.

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

## Data availability
There are no data to be shared.

## Code availability
The MATLAB codes used to prove Theorem 3 in the Supplementary Information can be found in the Supplementary Data.

**Acknowledgements** D.T. is a recipient of the DOC Fellowship OÖAS000047 of the Austrian Academy of Sciences at the Institute of Quantum Optics and Quantum Information (IQOQI), Vienna. M.W. and T.P.L. are each supported by a Lise Meitner Fellowship awarded by Fonds zur Förderung der wissenschaftlichen Forschung (FWF), with project numbers M 3109-N and M 2812-N, respectively. M.-O.R. and A.T. are supported by the Swiss National Fund Early Mobility Grants P2GEP2_191444 and P2GEP2 194800, respectively. We acknowledge support from the Government of Spain (FIS2020-TRANQI and Severo Ochoa CEX2019-000910-S), Fundacio Cellex, Fundacio Mir-Puig, Generalitat de Catalunya (CERCA, AGAUR SGR 1381 and QuantumCAT), the ERC AdG CERQUTE, the AXA Chair in Quantum Information Science and the Swiss NCCR SwissMap.

**Author contributions** All authors contributed equally to this work.

**Competing interests** The authors declare no competing interests.

**Additional information**
**Correspondence and requests for materials** should be addressed to Miguel Navascués.
