## [Peer Review File · Nature]

Reviewer Comments & Author Rebuttals

Reviewer Reports on the Initial Version:

Referee #1 (Remarks to the Author)

John Bell famously demonstrated the possibility of distinguishing empirically between quantum theory and an entire class of alternative theories, namely, the class of local realistic theories. This new paper similarly shows that there is an empirically detectable difference between quantum theory and a certain class of alternative theories. Roughly speaking, the distinction the authors establish is between quantum theory, which is based on the complex numbers, and a certain class of theories based on the real numbers. But what is this class of alternative theories, precisely?

It is *not* the class of theories whose formulations are free of complex numbers. This fact is pointed out in the paper's penultimate paragraph, where the authors cite examples of theories that do not require complex numbers and yet are empirically equivalent to quantum theory. In addition to the examples cited by the authors, there is also the classic paper by E. C. G. Stueckelberg, *Quantum Theory in Real Hilbert Space*, *Helv. Phys. Acta* **33**, 727 (1960). That paper shows how to construct a theory equivalent to quantum theory but based on a real Hilbert space.

If I understand the new paper correctly, the class of theories being distinguished from quantum theory consists of those theories that look exactly like ordinary quantum theory except that (i) the state of each system lies wholly in a real Hilbert space associated with that system and (ii) for any given system, the dimension of its real Hilbert space need not be the same as the dimension of the system's usual complex Hilbert space. Importantly, the theory must satisfy the following requirement, which I will call "requirement **T**" for "tensor product":

T: If \mathcal{H}_A is the real Hilbert space of system A, and \mathcal{H}_B is the real Hilbert space of system B, then the Hilbert space to be associated with the pair AB is $\mathcal{H}_A \otimes \mathcal{H}_B$, where the tensor product is over the real numbers.

Now, let us try to imagine a theory of this sort that we might reasonably hope would describe the actual world. In particular, let us focus on the *dimension* d of the real Hilbert space. One might have thought that a reasonable choice for a real quantum theory would be to make the dimension of the real Hilbert space always twice the dimension of the usual complex Hilbert space. Then the real vector space would be large enough to accommodate all the structure of the complex vector space. Indeed, this is a choice that might be suggested by the authors' Eq. (1). However, this choice conflicts with condition **T**. For example, if a qubit is represented in a 4-dimensional real space (twice the usual dimension), then by condition **T**, a pair of qubits would have to be represented in a 16-dimensional real space, which is not twice the usual dimension for such a system.

More generally, in constructing a quantum-like theory, we would normally take the dimension d to be a function only of the maximum number of states that can be reliably distinguished from each other when only a single copy of the system is available. Let us call this number n . In standard quantum theory, $d(n)$ is simply equal to n , but

we expect it to be larger in the real case. In addition, we would normally assume that when two systems A and B are considered jointly, n_{AB} is equal to the product $n_A n_B$. (One can distinguish exactly this many states by measuring the two systems separately.) Under these reasonable assumptions, condition **T** forces the function $d(n)$ to have this property:

$$d(n_A n_B) = d(n_A) d(n_B). \quad (1)$$

Functions satisfying Eq. (1) include functions of the form $d(n) = n^m$, where m is a positive integer. More generally, we could break n into its prime factors and replace each prime number in this decomposition with some (presumably larger) integer to get $d(n)$.

Finally, note that in any theory with $d \neq n$, there would have to be some structure in the theory accounting for the difference between d and n , that is, accounting for the fact that not all orthogonal states can be distinguished. This structure might take the form of superselection rules limiting the set of allowed observables.

We thus get a picture of the class of plausible theories the authors are distinguishing from standard quantum theory: they are real quantum theories satisfying condition **T**, with Hilbert space dimensions given by a function satisfying Eq. (1), and with additional structure accounting for the difference between d and n .

Now, one could argue against publishing this paper in *Nature* on the grounds that no physicists have tried seriously to develop the kind of theory that the authors hope to rule out. I, at least, am not aware of any papers that have wrestled with the details I have described in the preceding paragraphs: how does one choose the function $d(n)$ so that it satisfies Eq. (1), and what restrictions on observables might be needed to make the theory accord with observations? If researchers have not been exploring these questions, one might ask why we should consider it important to rule out this class of theories. But note that one could have made a similar objection to publishing Bell's theorem. Prior to Bell's work, no one had proposed a local realistic model that had a reasonable chance of accounting for all of the empirical predictions of quantum theory. The relevant question is not whether such theories have been worked out, but rather whether the class of theories the authors consider is a natural class, sufficiently plausible that it would be of significant interest to falsify them experimentally. On this question, I agree with the authors that the class they consider is a natural variant of quantum theory worthy of study. (The *most* natural variant of quantum theory in which the complex numbers are replaced with real numbers is the real quantum theory with $d = n$. But *this* theory can be ruled out by any number of elementary observations; one does not need the sophisticated methods of the paper under review.)

I think it is impressive that the authors have been able to distinguish the class of theories they are considering from standard quantum theory *without* knowing any details about, say, $d(n)$ or superselection rules. They have also done excellent work in showing that this distinction is accessible to experiment. The paper will surely inspire

experimentalists to try to achieve the proposed discrimination in the lab. Moreover, because it deals with the fundamental question of the role of complex numbers in physics, I think the paper would interest a broad audience. So I think the work is appropriate for publication in *Nature*.

However, I feel that the paper as currently written is much too likely to be misunderstood. A person reading, say, just the title and the first paragraph—or for that matter, all of the main text of the paper up to the last two paragraphs—could very easily get the impression that the empirical predictions of quantum theory cannot be captured by any theory that does not include complex numbers. In fact, one has to interpret the title and the opening paragraph extremely carefully to avoid drawing this conclusion. As the authors know, the empirical content of quantum theory can be accounted for by theoretical frameworks that do not include complex numbers. It is important to make sure the reader gets this message along with the very interesting and novel message that a certain natural *class* of real quantum theories cannot imitate the standard theory. When the kind of experiment that this work inspires is finally carried out successfully, it may be impossible to prevent inaccurate headlines in popular science articles—“Physicists prove imaginary numbers are real”—but we should make every effort to make the facts clear.

Primarily for this reason, I cannot recommend this paper for publication in its present form. I list here my specific concerns.

1. The statement made in the title—quantum physics needs complex numbers—can be interpreted in a way that makes it true, but the expression “quantum physics” has to be understood as entailing condition **T** if there are no complex numbers in the theory. I expect, though, that most readers would, on a first reading, take the title to be claiming that the *empirical content* of quantum physics requires complex numbers. Thus I think the title is misleading. A non-misleading statement would probably be too cumbersome to serve as a title. So I would recommend a title that is not a sentence, perhaps “The status of complex numbers in quantum physics”.
2. Throughout the paper, the authors use the adjective “quantum” as a shorthand to describe the kind of theoretical structure on which they focus their work. Consider, for example, this passage from the opening paragraph: “Thus, are complex numbers really necessary for a quantum description of nature? Here, we show this to be the case...” This statement can be regarded as true only if Stueckelberg’s formulation is not counted as a quantum description, but I expect most physicists would count it as a quantum description. Somehow, the authors need to find less ambiguous words to describe the kind of theory they are considering. They are considering descriptions of a specific form, with a real tensor product structure. Perhaps they could introduce a shorthand term for this kind of structure (other than simply “quantum”) and explain its meaning.

3. The most helpful thing the authors could do would be to state explicitly very near the beginning of the paper that the empirical content of quantum theory can be captured by theoretical frameworks not involving complex numbers. (Stueckelberg’s paper should be cited at that point.) Of course the authors could immediately go on to explain how the class of theories they are considering differs from these other frameworks.

4. There are a few other lines of research that the authors might consider citing in this connection. First, there is the classic paper by E. Wigner, *On the Quantum Correction For Thermodynamic Equilibrium*, Phys. Rev. **40**, 749 (1932). Eq. (11) of that paper expresses in entirely real terms the equation describing the evolution of the Wigner function, which itself is a real-valued function that can represent an arbitrary quantum state of an arbitrary number of particles. (Wigner does, however, use complex numbers to make the connection with the standard formulation.) Along different lines, it is possible in principle to express the empirical predictions of quantum theory in terms of ordinary probabilities, which are not only real but also non-negative. A paper developing this approach is Christopher A. Fuchs and Rüdiger Schack, *Quantum-Bayesian Coherence*, Rev. Mod. Phys. **85**, 1693 (2013). Finally, there is the more abstract mathematical perspective exemplified by John C. Baez, *Division Algebras and Quantum Theory*, Found. Phys. **42**, 819 (2012). Here is a quotation from the abstract of that paper: “. . . Hilbert spaces of any one of the three kinds—real, complex, and quaternionic—can be seen as Hilbert spaces of the other kinds equipped with extra structure.” Baez refers to a paper by Freeman Dyson, who argued that, as regards representations of symmetries in quantum theory, it is actually *preferable* to base the formalism on the real numbers rather than the complex numbers.

5. On page 2, the authors write, “Note that if an experiment does not violate any Bell inequality, it is possible to reproduce the measured probabilities using a classical local deterministic model and, as said, real numbers suffice for classical physics. A Bell violation is therefore a necessary condition for a possible complex-real quantum gap.” I think the word “classical” is being used in two distinct senses here. The reference to what had been said earlier refers to the actual theories of classical physics such as Maxwell’s electrodynamics. But a local deterministic model that might explain some non-Bell-violating experiment need not be one of these familiar theories. So more needs to be said in order to reach the conclusion that a Bell violation is a necessary condition.

I would be happy to review a revised version of the manuscript that addresses these concerns.

Referee #2 (Remarks to the Author):

The main result of the manuscript is that certain quantum correlations involving three parties sharing two independent sources cannot be reproduced by using only quantum states and measurements with real coefficients – however large the dimension of the Hilbert spaces used.

The result is certainly very interesting for a physics audience, however, the title claim is overstated: it is in fact possible to reproduce all the quantum correlations, including those involved in the experiment proposed here, without using complex numbers. The authors acknowledge this in the second-last paragraph. What the authors show is that theories that aim to do that must necessarily violate other assumptions -- which are implicit in the present work -- such as the rules for composition of systems and the calculus of probabilities. However, it is not entirely surprising that modifying some aspect of the quantum formalism would necessarily require modifying some other aspects, if one wishes to maintain the same empirical predictions.

More specifically, what is ruled out are real quantum theories where independently-prepared and measured systems are represented by independent real quantum states and measurements described as tensor products (plus some further extra assumptions about causal structure, the Born rule, and possibly others). Although this sounds reasonable, when we look at a construction like Eq. (1), it seems clear that one ought not to read it explicitly as a mixture of states of two parallel-composed systems, as it would a priori be interpreted in ordinary complex QM: the second term in the mixture contains an operator that is not a valid quantum state in general, and it should be clear that the auxiliary qubit cannot be independently addressed as a system in its own right, but is only playing a mathematical role. This already indicates that something about the composition of systems in ordinary quantum theory may not in general carry over to a real Hilbert space construction. The fact that the real-Hilbert space construction of ref. [33] contains an “universal rebit” (that cannot be thought of as localised in space but is somehow shared by all systems) may be seen as further evidence that such constructions would likely violate the usual composition rules in one way or another. The main novelty here is to prove that this is indeed necessary.

The paper would thus be more useful if written for physicists, without the long introduction about complex numbers, which is as argued above somewhat misleading anyway. In such a rewritten version, it should start by reviewing the previous arguments for ruling out real Hilbert spaces e.g. local tomography, previous explicit constructions such as ref [33] (and more usefully going back to Stueckelberg, etc), and explaining what is offered here: a device-independent reason, with an assumption about the causal structure of a network underlying the experiment, as reflected in the tensor-product structure used to define independent sources. These assumptions should be made explicit, to clearly distinguish what is ruled out here from apparent counterexamples such as that of ref [33]. It would also be much more useful if at least a sketch of the proof were in the main text, perhaps in a Methods section, rather than buried in the SI.

More than ruling out real quantum theory, I take the real novelty of the work to be that it provides a very interesting demonstration that constraints from causal structure in network configurations beyond the standard Bell scenarios can bring out novel quantum features in device-independent scenarios. I think this aspect should be much more emphasised and its potential implications further delineated.

I would thus not recommend publication in Nature, though I believe it could be potentially appropriate for a lower-tier Nature journal such as Nature Communications, after the suggested revisions above, and after the authors satisfactorily address the following further comments (some of which regarding the correctness of the proofs):

1) Eq (1) is confusing for a non-expert reader, since it explicitly involves complex numbers and operations. Of course, the state is real because $\rho = \rho^*$, but a reader who is not even familiar

with complex numbers (as is the target readership, given the introduction), will be puzzled by it. Perhaps the authors could show with an explicit example that a state of this form can be written without using i or $*$ at all, in terms of the real and imaginary parts of the coefficients of ρ .

2) In the last paragraph, is [34] really the paper that started the stated research program? It sounds like one variant of the research program that has been going for decades to derive quantum correlations from principles.

Regarding the SI:

3) If you're going to allow for shared randomness, that must mean that the sources have a common cause. But in that case, since you are in a device-independent scenario, why wouldn't you also allow for shared entanglement, or for correlated measurement operators?

4) I suppose it also makes no difference to have the measurement choices being classically correlated, as long as they are not correlated with the states? If so, it could be useful to also state it explicitly.

5) Page 1 of the SI, the authors state "This state does not admit a real separable decomposition... This can be seen by tracing the first part of the state and noticing that the resulting state..." But why would we trace the first part out? When I read a state of the form (1), I assume that this is just a real representation of the original state, rather than taking too seriously the auxiliary subsystem as something that has its own marginal quantum state, which can be obtained by partial trace. Even if we want to assume that both Alice's state and Bob's state are independently real (which is related to the main assumption about independent systems used in this work), the reasonable assumption here would be that this is so when taking into account their respective auxiliary systems, which should be read as being an inseparable part of each system.

6) Page 3: the authors consider a real purification, stating that "at the end of the day, we will trace out the purifying system". But it is not clear to me where this happens, or how it's guaranteed that Charlie's operators act trivially on the purifying system. This is pretty important, since the standard real-state construction isn't a pure state to start with – so a purification would probably be non-physical in general in a theory of this type. It must be made crystal clear that this is not affecting the validity of the proof.

7) Is there an implicit assumption about the dynamics of candidate theories built into the use of the isometries in the proof of Th. 1? If in the standard real construction the measurement operators need to be modified to act on the auxiliary systems, shouldn't a similar thing occur in dynamics? How is this enforced here?

8) Below Eq.(6): I don't think that shows that one needs to prepare that state for Alice and Charlie "as the construction requires"; only that one needs to prepare some state that can be converted into that state after an appropriate isometry and tracing out the systems Alice and Charlie started with.

9) In the final step of the proof of Theorem 1, the contradiction is claimed to be reached by pointing out that the state in systems $A''C''$ is not real separable. But I am not entirely convinced that this settles the proof beyond doubt. Firstly, as pointed out in comment 5) above, I don't think this would be the relevant question if $A''C''$ were the original auxiliary rebits. If on the other hand (9) were a state of the original system, one could argue that it is not real separable across the A''/C'' partition. However, this is so after taking the mixture of the four b outcomes, and it is not clear to me whether the purification assumption may have introduced an error here.

Furthermore, there is the question whether the isometries considered were too general, and whether a real quantum theory with restricted dynamics could block the possibility to prepare state (9) by disallowing the required isometries (while nevertheless still reproducing the required quantum correlations). Ideally the proof should not use any dynamics, if it wants to rule out the most general class of theories.

Note that this point may be less important if the result is seen as a device-independent proof within standard quantum theory rather than ruling out alternatives to quantum theory, which could be another reason to motivate it as such.

Note also that much of these comments carry over to the proofs of Thms 2 and 3.

Minor comments/corrections:

- a) It would be useful to have a subscript in the $|b\rangle\langle b|$ factor in Eq.(6) of SI.
- b) Superscript typo in (8) of SI.

Referee #3 (Remarks to the Author):

This paper reports on a set of correlations that can be generated in quantum theory, but which could not be generated in the analog of quantum theory with real density matrices and real measurements (called real quantum mechanics in the foundations literature). The paper shows that there is an experiment that can rule out real quantum mechanics. There are already good theoretical reasons for rejecting real quantum mechanics (e.g. the failure of local tomography) and it is ruled out by several axiomatic approaches to quantum theory e.g. quant-ph/0101012 or Phys. Rev. A 84, 012311. Nevertheless having an experiment that can directly eliminate it is an advantage. The result can be thought of analogously to Bell's work: Bell's work rules out local classical theories, while this paper rules out real quantum mechanics.

Although this is a nice result, the paper significantly oversells it. This starts with the title, which taken literally is false, and has already prompted a clarification: arXiv:2103.12740. Furthermore, the abstract says that real quantum physics can reproduce the outcomes of any multipartite experiment, again in contradiction with the title (unless the claim is that the works referred to are wrong). The point is that these previous works use real states that do not respect the usual tensor product structure and are "nonlocal" in a way that ordinary quantum theory is not. This is mentioned in the second to last paragraph of the present paper and this paragraph gives a fair statement of the result (and contradicts other statements in the paper).

Therefore the paper needs significant rewriting to avoid misleading readers. The paper should be rephrased stating that it gives a proposed experimental test that rules out real quantum mechanics (it is much better to talk about what is ruled out since theories can be falsified but not verified). Claims that complex numbers are necessary (without further qualification) should be removed. Without such a rewriting, the results of this work will be misunderstood and I cannot support publication.

While I think the main result (when correctly) stated is novel and interesting, but may not be of broad enough appeal to warrant publication in Nature.

Minor remarks:

The authors define CHSH(1,2;1,2) but then use CHSH(1,3;3,4) etc. it would be clearer to define CHSH(x1,x2;y1,y2).

In the supplementary:

Comment on why you have projective measurements

When you say "any state shared between Alice and Charlie after tracing out Bob...", are you postselecting on Bob's outcomes? I didn't understand exactly what you were referring to. Maybe write down the state to make this clear. Also, what "quantum operations" are you referring to?

A link is then made to Fig.1, but the fig has Alice and Bob and not Charlie. Are you imagining a different scenario where Charlie is like Bob in Fig.1 and they just get the state without a third

party measuring?

Explain the link between (4) and the variants of CHSH₃, i.e., say which variant of CHSH₃ is played depending on b .

The text is unclear about whether there is a restriction on dimension or not. I think the authors allow the states in real quantum mechanics to have arbitrary (perhaps finite) dimension. If so, this should be stressed. The operators $D^C_{\{zx\}}$ etc. are not defined using (3).

The "regularized versions of X^C " need to be explained

Author Rebuttals to Initial Comments:

Reply to referee #1:

First of all, we thank the referee for the time she/he dedicated to review our manuscript, and the very clear formulation of her/his interpretation of our result. We are very pleased to read that the referee thinks that our “work is appropriate for publication in Nature”, provided that their comments are adequately accounted for.

Before coming to a detailed reply, we like to start with some general comments. Obviously, any mathematical model involving complex numbers can be turned into a “real” model with the same computational complexity and predictive power just by replacing every instance of a complex number by a pair of real numbers. The question of whether a model can be “realified” is therefore non-trivial only when one fixes the theoretical framework. In our paper, we consider the standard Hilbert space with the bra-ket framework of quantum theory (defined below), and we wonder whether one can explain the same experiments when we impose that the basic elements of the framework, namely, the bras and the kets, are all real. As the referee correctly pointed out, one can also describe quantum mechanics experiments via (real) Wigner functions, or in terms of (real) probability vectors. Those two perfectly valid frameworks are, however, different from the standard bra-ket framework that we consider in our work.

More specifically, the bra-ket quantum framework (dubbed “quantum theory” in the new version of our manuscript) satisfies the following postulates:

- (i) To every physical system S , there corresponds a complex Hilbert space H_S and its state is represented by a normalized vector ϕ in H_S , that is, $\langle \phi | \phi \rangle = 1$.
- (ii) A measurement Π in S corresponds to an ensemble $\{\Pi_r\}_r$ of projection operators acting on H_S , with $\sum_r \Pi_r = 1$.
- (iii) The Born rule: If we measure Π when system S is in state ϕ , the probability of obtaining result r is given by $Pr(r) = \langle \phi | \Pi_r | \phi \rangle$.
- (iv) The Hilbert space H_{ST} corresponding to the composition of two systems S, T is $H_S \otimes H_T$. The operators used to describe measurements or transformations in system S act trivially on H_T and vice-versa. Similarly, the state representing two independent preparations of the two systems is the tensor product of the two preparations.

All these postulates (including (iv)) can be found in John von Neumann’s “The Mathematical foundations of Quantum mechanics” and in Paul Dirac’s “Principles of Quantum Mechanics”, as well as in modern textbooks on quantum physics (e.g.: Nielsen and Chuang’s). They constitute the definition of quantum theory; any mathematical model violating any of the above postulates cannot be properly regarded as “quantum theory”.

It is true that, for those areas of physics concerned with the observations of a single system, one can do without the last postulate. However, quantum theory, as originally introduced, is a multi-system theory, where postulate (iv) is used to define system composition.

In our paper, we consider an alternative theory, “real quantum theory”, where postulate (i) is replaced by

- (i’) To every physical system S , there corresponds a real Hilbert space H_S and its state is represented by a normalized vector ϕ in H_S , that is, $\langle \phi | \phi \rangle = 1$.

This theory is formally equivalent to postulates (i)-(iv), with the restriction that all states and operators have real entries.

There exist infinitely many models that fit the framework of real quantum theory. Some of such models satisfy some super-selection rules, but others don't. Our result invalidates them all: no theory satisfying postulates (i'), (ii)-(iv), constrained or not, can explain the statistics observed in the proposed quantum tripartite experiment with two independent sources of entangled states.

In his seminal paper on quantum theory with real numbers, Stueckelberg defines a real mathematical framework to describe single-site quantum experiments. At no point does Stueckelberg consider experiments involving multiple systems, such as a Bell experiment or a demonstration of quantum teleportation. Not surprisingly, a straightforward generalization of Stueckelberg's theory to the multipartite case (the universal real qubit) does not satisfy postulate (iv) *and thus is not quantum theory*. This should come as no surprise, because, in Stueckelberg's times, entanglement and multi-partite experiments were barely discussed. Entanglement, the central piece of quantum physics, was recognized only with the emergence of quantum information science in the 1990's. Our result implies that no generalization of Stueckelberg's theory will, at the same time, satisfy postulates (i'),(ii)-(iv) and lead to the same predictions as standard quantum theory.

The first version of our manuscript could indeed be misleading, our initial title and introduction could give the impression that we proved that the empirical content of quantum physics requires complex numbers. As we already explained in the conclusion of our first version, this is obviously not the case. In our rewriting, our first concern was to avoid any misleading statement in this direction. We also added two appendices presenting a more elaborated discussion on this key point.

Let us now come to a detailed reply to the referee's report.

Roughly speaking, the distinction the authors establish is between quantum theory, which is based on the complex numbers, and a certain class of theories based on the real numbers. But what is this class of alternative theories, precisely?

Indeed, this was not sufficiently clear in the previous version. In the revised manuscript, we clarify exactly which class of theories we refute. Note that we do not only consider models that assign to each complex quantum physical system S a real quantum counterpart S' , equipped with the appropriate super-selection rules to ensure that S' *simulates* the theoretical behavior of S . Instead, we consider any model satisfying postulates (i'), (ii)-(iv).

More generally, in constructing a quantum-like theory, we would normally take the dimension d to be a function only of the maximum number of states that can be reliably distinguished from each other when only a single copy of the system is available. Let us call this number n .

[...]

The most natural variant of quantum theory in which the complex numbers are replaced with real numbers is the real quantum theory with $d = n$. But this theory can be ruled out by any number of elementary

observations; one does not need the sophisticated methods of the paper under review.)

The referee is referring to a theory where all states and measurements compatible with postulates (i'), (ii)-(iv) are allowed. Provided that one has a promise on the Hilbert space dimension of the system under observation, one can indeed devise simple quantum experiments to falsify such a “unrestricted” real quantum theory. Note, however, that Hilbert space dimension cannot be experimentally upper bounded, making such an assumption fundamentally not desirable. Interpreting such experiments as a disproof of real quantum theory thus requires a considerable leap of faith.

It is important to make sure the reader gets this message along with the very interesting and novel message that a certain natural class of real quantum theories cannot imitate the standard theory. When the kind of experiment that this work inspires is finally carried out successfully, it may be impossible to prevent inaccurate headlines in popular science articles “Physicists prove imaginary numbers are real” but we should make every effort to make the facts clear. [...] I expect, though, that most readers would, on a first reading, take the title to be claiming that the empirical content of quantum physics requires complex numbers. Thus I think the title is misleading. A nonmisleading statement would probably be too cumbersome to serve as a title. So I would recommend a title that is not a sentence, perhaps “The status of complex numbers in quantum physics”.

We thank the referee for these important warnings on the potential misunderstanding of our work. We completely agree with the referee and have modified the title, abstract and introduction accordingly. The new title reads: “quantum theory needs complex numbers”. The idea that we want to transmit is that the elements of the bra-ket framework cannot be taken real without the framework changing part of its predictive power. By adopting the term ‘quantum theory’, we refer to the standard formulation of quantum theory in Hilbert space, as opposed to the many different formulations that are accommodated under the roof of ‘quantum physics’. The first sentence of the abstract makes it clear that the empirical content of quantum theory can be captured without complex numbers, and the following sentences give a succinct but clear intuitive definition of the signification of our shorthand term “quantum theory”. We also explain this point in the new introduction. As this term is used throughout our paper, we believe our title present our work in a well-balanced way, and that our work makes it very clear that it does not claim that “[we] prove imaginary numbers are real”.

Throughout the paper, the authors use the adjective “quantum” as a shorthand to describe the kind of theoretical structure on which they focus their work. They are considering descriptions of a specific form, with a real tensor product structure. Perhaps they could introduce a shorthand term for this kind of structure (other than simply “quantum”) and explain its meaning.

The referee is right. To remedy this, the new version of the paper explicitly states the postulates of what we call quantum and real quantum theory. We avoided any unclear use of the adjective “quantum” alone.

The most helpful thing the authors could do would be to state explicitly very near the beginning of the paper that the empirical content of quantum theory can be captured by theoretical frameworks not involving complex numbers. (Stueckelberg's paper should be cited at that point.)

We followed the referee's advice: in the new version, we explain that "realifying" a mathematical model becomes non-trivial only when we enforce a mathematical structure. Right afterwards, we introduce the structure that we consider, namely, the postulates of quantum theory. We mention Stueckelberg's paper a bit later, after the introduction of the postulates of quantum theory, for clarity.

There are a few other lines of research that the authors might consider citing in this connection. [...]

We now make a clear reference to these extra works (description of quantum mechanics experiments via (real) Wigner functions, or in terms of (real) probability vectors).

I think the word "classical" is being used in two distinct senses here.

Indeed. When we speak of Bell inequalities, our notion of "classical" is much more inclusive than when we speak of, say, electromagnetism: in the first case, we mean any local realistic physical theory. Since it does not play an important role in our paper, due to space constraints, we opted for keeping the term. Although not rigorous, this shorthand terminology is standard since many years and appears e.g. in the published articles of the 2015 loophole-free Bell experiments.

Reply to referee #2:

First of all, we thank the referee for the time she/he dedicated to review our manuscript. We are pleased to see that the referee finds our results very interesting. We thank her/him for the criticisms expressed in the report, which made us understand that the presentation of our work could be misleading, and required more clarity.

Before coming to a detailed reply, we like to start with some general comments. Obviously, any mathematical model involving complex numbers can be turned into a “real” model with the same computational complexity and predictive power just by replacing every instance of a complex number by a pair of real numbers. The question of whether a model can be “realified” is therefore non-trivial only when one fixes the theoretical framework. In our paper, we consider the standard Hilbert space with the bra-ket framework of quantum theory (defined below), and we wonder whether one can explain the same experiments when we impose that the basic elements of the framework, namely, the bras and the kets, are all real. One can also describe quantum mechanics experiments via (real) Wigner functions, or in terms of (real) probability vectors. Those two perfectly valid frameworks are, however, different from the standard bra-ket framework that we consider in our work.

More specifically, the bra-ket quantum framework (dubbed “quantum theory” in the new version of our manuscript) satisfies the following postulates:

- (i) To every physical system S , there corresponds a complex Hilbert space H_S and its state is represented by a normalized vector ϕ in H_S , that is, $\langle \phi | \phi \rangle = 1$.
- (ii) A measurement Π in S corresponds to an ensemble $\{\Pi_r\}_r$ of projection operators acting on H_S , with $\sum_r \Pi_r = 1$.
- (iii) The Born rule: If we measure Π when system S is in state ϕ , the probability of obtaining result r is given by $Pr(r) = \langle \phi | \Pi_r | \phi \rangle$.
- (iv) The Hilbert space H_{ST} corresponding to the composition of two systems S, T is $H_S \otimes H_T$. The operators used to describe measurements or transformations in system S act trivially on H_T and vice-versa. Similarly, the state representing two independent preparations of the two systems is the tensor product of the two preparations.

All these postulates (including (iv)) can be found in John von Neumann’s “The Mathematical foundations of Quantum mechanics” and in Paul Dirac’s “Principles of Quantum Mechanics”, as well as in modern textbooks on quantum physics (e.g.: Nielsen and Chuang’s). They constitute the definition of quantum theory; any mathematical model violating any of the above postulates cannot be properly regarded as “quantum theory”.

It is true that, for those areas of physics concerned with the observations of a single system, one can do without the last postulate. However, quantum theory, as originally introduced, is a multi-system theory, where postulate (iv) is used to define system composition.

In our paper, we consider an alternative theory, “real quantum theory”, where postulate (i) is replaced by

- (i’) To every physical system S , there corresponds a real Hilbert space H_S and its state is represented by a normalized vector ϕ in H_S , that is, $\langle \phi | \phi \rangle = 1$.

This theory is formally equivalent to postulates (i)-(iv), with the restriction that all states and operators have real entries.

There exist infinitely many models that fit the framework of real quantum theory. Some of such models satisfy some super-selection rules, like the ones discussed in the referee's report, but others don't. Our result invalidates them all: no theory satisfying postulates (i'), (ii)-(iv), constrained or not, can explain the statistics observed in the proposed quantum tripartite experiment with two independent sources of entangled states.

In his seminal paper on quantum theory with real numbers, Stueckelberg defines a real mathematical framework to describe single-site quantum experiments. At no point does Stueckelberg consider experiments involving multiple systems, such as a Bell experiment or a demonstration of quantum teleportation. Not surprisingly, a straightforward generalization of Stueckelberg's theory to the multipartite case (the universal real qubit) does not satisfy postulate (iv) *and thus is not quantum theory*. This should come as no surprise, because, in Stueckelberg's times, entanglement and multi-partite experiments were barely discussed. Entanglement, the central piece of quantum physics, was recognized only with the emergence of quantum information science in the 1990's. Our result implies that no generalization of Stueckelberg's theory will, at the same time, satisfy postulates (i'), (ii)-(iv) and lead to the same predictions as standard quantum theory.

We now present a detailed reply to the referee's report.

The title claim is overstated: it is in fact possible to reproduce all the quantum correlations, including those involved in the experiment proposed here, without using complex numbers. The authors acknowledge this in the second-last paragraph. What the authors show is that theories that aim to do that must necessarily violate other assumptions -- which are implicit in the present work -- such as the rules for composition of systems and the calculus of probabilities.

We agree with the referee, in fact our title, abstract and introduction did not make it clear enough that our result does not imply that there do not exist real theories which capture the empirical content of quantum physics without complex numbers. We now made this very explicit. We also made explicit the assumptions, or postulates, of quantum theory.

It is not entirely surprising that modifying some aspect of the quantum formalism would necessarily require modifying some other aspects, if one wishes to maintain the same empirical predictions.

This remark concerns our main result: in quantum theory, going from complex numbers (postulate (i)) to real ones (postulate (i')) cannot maintain the same empirical predictions if one assumes the existence of independent sources and measurements, as is very natural in quantum networks.

Indeed, most often a modification of a physical theory results in a change in its empirical predictions, and requires a second correction to keep these predictions unchanged. In other words, an uncorrected modification of a physical theory often leads to different predictions, hence is falsifiable by an experiment.

However, we don't completely agree with the referee when they write that, in this particular case, our result is not surprising. In fact, the measurement statistics of any multipartite quantum Bell experiment *can be reproduced with real quantum systems*. Concretely, going from postulate (i) to postulate (i') does not require modifying any other aspect of the theory to maintain the same empirical predictions in the standard Bell scenario, with one source connected to all parties. The same applies to scenarios involving joint measurements on states prepared by different sources, as in the PBR theorem.

In view of these results, it was generally believed in the quantum foundations community that real quantum theory was experimentally undistinguishable from complex quantum theory. Even some of us tried (and of course failed) to prove this result for general networks some time ago. Our result was therefore a surprise, at least to us and many colleagues. Finally, whether a result is "surprising" is a rather subjective argument that varies among researchers. A less subjective criterion is whether the results are relevant. We are convinced that our results are as they demonstrate the need of complex numbers in quantum theory. Our belief is also based on the very positive feedback received by our colleagues and the many received invitations to explain our findings in conferences and seminars.

More specifically, what is ruled out are real quantum theories where independently-prepared and measured systems are represented by independent real quantum states and measurements described as tensor products (plus some further extra assumptions about causal structure, the Born rule, and possibly others).

We now clarified the presentation of our main result. The postulates presented in our work are not extra assumptions; rather, they constitute the very definition of quantum theory, as originally conceived by Dirac and von Neumann. One can also find these principles in modern introductions to quantum physics (e.g.: Nielsen and Chuang's "Quantum Computation and Quantum Information").

it seems clear that one ought not to read it explicitly as a mixture of states of two parallel-composed systems, as it would a priori be interpreted in ordinary complex QM: the second term in the mixture contains an operator that is not a valid quantum state in general, and it should be clear that the auxiliary qubit cannot be independently addressed as a system in its own right, but is only playing a mathematical role.

Please note that the specific real construction presented in Figure 1 is only there to explain that the measurement statistics of any multipartite quantum Bell experiment can be reproduced with real quantum systems. It does not imply that the states in the considered real quantum theory need to have that form. Our main results, rigorously stated in section D (section A, in the previous version) of the Supplementary Information in the form of Propositions 1 and 2 and Theorem 3, make no reference to this specific real construction, or to any super-selection rules and correspondences between complex and real quantum systems. We added a new section C in the Supplementary Information which explains how the measurement statistics of a joint measurement on independent preparations can be reproduced with real quantum systems: note that in this case, the construction is completely different. Any ontologic consideration on the reality of these different constructions is beyond the scope of our work. They simply prove that complex and real quantum theories have the same predictive power in the considered experimental scenarios.

The paper would thus be more useful if written for physicists, without the long introduction about complex numbers, which is as argued above somewhat misleading anyway. In such a rewritten version, it should start by reviewing the previous arguments for ruling out real Hilbert spaces e.g. local tomography, previous explicit constructions such as ref [33] (and more usefully going back to Stueckelberg, etc), and explaining what is offered here: a device-independent reason, with an assumption about the causal structure of a network underlying the experiment, as reflected in the tensor-product structure used to define independent sources. These assumptions should be made explicit, to clearly distinguish what is ruled out here from apparent counterexamples such as that of ref [33]. It would also be much more useful if at least a sketch of the proof were in the main text, perhaps in a Methods section, rather than buried in the SI.

More than ruling out real quantum theory, I take the real novelty of the work to be that it provides a very interesting demonstration that constraints from causal structure in network configurations beyond the standard Bell scenarios can bring out novel quantum features in device-independent scenarios. I think this aspect should be much more emphasised and its potential implications further delineated.

We disagree with the assessment on the real novelty of our work, again a statement that contains some degree of subjectivity. In our view, the real novelty of our work is to prove, for the first time, that replacing complex by real numbers in quantum theory has experimental consequences. We agree with the referee that causal constraints play a fundamental role in the derivation of the main result, but this is a novelty more on the methods side, while the gap between real and quantum theory is more on the conceptual and fundamental side.

We however agree with the referee that the presentation of our results could be improved and hope that our new version makes this clear. In preparing it, we tried to accommodate all the remarks in the three reports, which were very valuable, within the space constraints.

The comment by the referee on the importance of causal constraints also motivated us to consider PBR geometries and prove that also there real and complex quantum theory are experimentally equivalent. We also include in the main text the sentence “Our results demonstrate how these networks, beyond their practical relevance, open radically new perspectives to solve open questions in the foundations of quantum theory when exploiting the causal constraints associated to their geometries”.

1) Eq (1) is confusing for a non-expert reader, since it explicitly involves complex numbers and operations. Of course, the state is real because $\rho = \rho^$, but a reader who is not even familiar with complex numbers (as is the target readership, given the introduction), will be puzzled by it. Perhaps the authors could show with an explicit example that a state of this form can be written without using ‘i’ or ‘*’ at all, in terms of the real and imaginary parts of the coefficients of ρ .*

We have followed the referee’s advice and expressed $\tilde{\rho}$ in terms of the real and imaginary parts of ρ .

2) In the last paragraph, is [34] really the paper that started the stated research program? It sounds like one variant of the research program that has been going for decades to derive quantum correlations from principles.

A relatively old research program consists of, starting from the set of no-signalling correlations, adding physical postulates until the set of quantum correlations is recovered. [34] introduces a new program which aims to single out the quantum set without postulates, just by identifying a given task where the set of quantum correlations performs maximally.

3) If you're going to allow for shared randomness, that must mean that the sources have a common cause. But in that case, since you are in a device-independent scenario, why wouldn't you also allow for shared entanglement, or for correlated measurement operators?

Of course, our proof also applies in the absence of shared randomness. If we consider the scenario of shared randomness is because of the reasons explained in Section H (previously E) of the Supplementary Information (e.g.: robustness under shared randomness allows us to devise an experiment to disprove real quantum theory that does not rely on the i.i.d. assumption between experimental rounds). In addition, shared randomness is expected to occur, e.g., if one uses a classical device to synchronize the two sources needed to conduct the experiment, or if Alice, Bob and Charlie exchange information about their outcomes between experimental rounds. Classically correlated measurement operators can also be modeled through shared randomness (by making the measurement operator at each site depend on the value of the classical variable), so our model is quite general.

Admittedly, we cannot allow shared entanglement between the sources; otherwise we would be in a standard Bell scenario, where there is (provenly) no separation between real and complex quantum correlations. As we explain in the main text, this is the natural and common way to describe the setup we are considering.

4) I suppose it also makes no difference to have the measurement choices being classically correlated, as long as they are not correlated with the states? If so, it could be useful to also state it explicitly.

Indeed, we can, for instance, think of our setup as a non-local game, in a similar way as we often do in a Bell scenario. This means that the measurement settings can be thought of as questions that are provided by a referee and the outcomes as the players' answers. The referee is then in principle free to choose the questions he asks in a correlated way.

Notice that it is, however, important that the measurement setting that each party receives is not only uncorrelated with the states but also unknown to the other players. Otherwise, the players could generate any correlations they want, even classically.

This said, for the experiment we propose, letting the players independently sample their settings seems more natural to us than introducing a source of shared randomness that the players can use to generate their settings but not their outcomes.

5) Page 1 of the SI, the authors state “This state does not admit a real separable decomposition... This can be seen by tracing the first part of the state and noticing that the resulting state...” But why would we trace the first part out? When I read a state of the form (1), I assume that this is just a real representation of the original state, rather than taking too seriously the auxiliary subsystem as something that has its own marginal quantum state

The goal was to show that the specific real construction sketched in the main text does not lead to a real quantum description of the swap scenario, in the sense of the second formula of Figure 2. In that formula, the state shared by Alice and Charlie at the beginning of the experiment must be real separable. The state prescribed by the real construction is, however, real entangled. Hence, the real construction in the text does not provide a real quantum representation for the measurement statistics (in the sense of the second formula of Figure 2).

This argument does not rely on any ontological interpretation that one might assign to the real state generated by the real construction: it follows from the mathematical definition of real separability and the property that real separable states remain separable after any local real operation, such as a partial trace.

6) Page 3: the authors consider a real purification, stating that “at the end of the day, we will trace out the purifying system”. But it is not clear to me where this happens, or how it’s guaranteed that Charlie’s operators act trivially on the purifying system. This is pretty important, since the standard real-state construction isn’t a pure state to start with – so a purification would probably be non-physical in general in a theory of this type. It must be made crystal clear that this is not affecting the validity of the proof.

It is guaranteed because we are postulating where Charlie’s measurements act: in the original system C. The purification system that we assign to Charlie does not play any role in the proof, as it is not acted upon by any operator (note that the two isometries are constructed from Alice and Charlie’s operators, which by definition only act non-trivially on A and the original system C). Moreover, the purification system is traced out, together with system A and the original system C, after the action of the isometry. In any case, we expanded the explanation in the SI; this should hopefully clarify the matter.

7) Is there an implicit assumption about the dynamics of candidate theories built into the use of the isometries in the proof of Th. 1? If in the standard real construction the measurement operators need to be modified to act on the auxiliary systems, shouldn’t a similar thing occur in dynamics? How is this enforced here?

In the proof of Th.1, we do not consider any specific real construction. Our proof is simply based on the postulates (i’, ii-iv), the independence of the sources and measurements, and some observed statistics in an experiment: it does not contain any other implicit assumption. The theory does not include any dynamical content.

8) Below Eq.(6): I don't think that shows that one needs to prepare that state for Alice and Charlie "as the construction requires"; only that one needs to prepare some state that can be converted into that state after an appropriate isometry and tracing out the systems Alice and Charlie started with.

We clarified this point, adding a comment making explicit that the state is prepared up to local isometries.

9) In the final step of the proof of Theorem 1, the contradiction is claimed to be reached by pointing out that the state in systems $A''C''$ is not real separable. But I am not entirely convinced that this settles the proof beyond doubt. Firstly, as pointed out in comment 5) above, I don't think this would be the relevant question if $A''C''$ were the original auxiliary rebits. If on the other hand (9) were a state of the original system, one could argue that it is not real separable across the $A'A''/C'C''$ partition. However, this is so after taking the mixture of the four b outcomes, and it is not clear to me whether the purification assumption may have introduced an error here. Furthermore, there is the question whether the isometries considered were too general, and whether a real quantum theory with restricted dynamics could block the possibility to prepare state (9) by disallowing the required isometries (while nevertheless still reproducing the required quantum correlations). Ideally the proof should not use any dynamics, if it wants to rule out the most general class of theories.

The goal of that section is to prove Propositions 1 and 2, which state that the statistics \bar{P} cannot be approximated arbitrarily well by distributions of the form

$$P(a, b, c|x, z) = \sum_{\lambda} P(\lambda) \text{tr}\{(\tilde{\sigma}_{AB_1}^{\lambda} \otimes \tilde{\sigma}_{B_2C}^{\lambda})(\tilde{A}_{a|x} \otimes \tilde{B}_b \otimes \tilde{C}_{c|z})\},$$

where $\tilde{\sigma}_{AB_1}^{\lambda}, \tilde{\sigma}_{B_2C}^{\lambda}$ are real quantum states and $\tilde{A}_{a|x}, \tilde{B}_b, \tilde{C}_{c|z}$ are real measurement operators acting on arbitrary Hilbert spaces A, B_1, B_2, C . The proof proceeds by contradiction: if such real states and measurement operators exist, then one can use the measurement operators to define some local isometries that, applied to the systems A, C , generate a real entangled state. This is impossible, since systems A and C are in a real separable state at the beginning of the experiment.

This proof should be regarded as a mathematical proof, based on the postulates (i', ii-iv), the independence of the sources, and the observed statistics \bar{P} (which are predicted to exist according to quantum theory). Real separability is only a mathematical property, deduced from postulate (iv), which leads to a (mathematical) contradiction.

As explained in our answer to point 6) and clarified in our SI, the purification is not an additional assumption, and does not introduce errors.

Our proof involves no dynamics. The isometries are mathematical objects which are proven to exist and lead to a mathematical contradiction, but there is no necessity of "applying them to the state in practice, in a dynamic way". In particular, a theory which would restrict dynamics is also rejected by our result: the

fact that one can mathematically define some real local isometries is independent from the question of whether such isometries can be physically implemented in the hypothetical real quantum theory generating the said correlations.

Minor comments/corrections:

a) It would be useful to have a subscript in the $|b\rangle\langle b|$ factor in Eq.(6) of SI.

b) Superscript typo in (8) of SI.

We thank the referee for pointing these out. We have adapted these as requested.

Reply to referee #3:

First of all, we thank the referee for the time she/he dedicated to review our manuscript.

While I think the main result (when correctly) stated is novel and interesting, but may not be of broad enough appeal to warrant publication in Nature.

We are pleased to read that the referee finds our main result nice, novel and interesting. We hope that the new version of the paper, and our arguments in this reply, convince her/him of its importance for quantum physics and recommend its acceptance.

Before coming to a detailed reply, we like to start with some general comments. Obviously, any mathematical model involving complex numbers can be turned into a “real” model with the same computational complexity and predictive power just by replacing every instance of a complex number by a pair of real numbers. The question of whether a model can be “realified” is therefore non-trivial only when one fixes the theoretical framework. In our paper, we consider the standard Hilbert space with the bra-ket framework of quantum theory (defined below), and we wonder whether one can explain the same experiments when we impose that the basic elements of the framework, namely, the bras and the kets, are all real. Admittedly, one can also describe quantum mechanics experiments via (real) Wigner functions, or in terms of (real) probability vectors. Those two perfectly valid frameworks are, however, different from the standard bra-ket framework that we consider in our work.

More specifically, the bra-ket quantum framework (dubbed “quantum theory” in the new version of our manuscript) satisfies the following postulates:

- (i) To every physical system S , there corresponds a complex Hilbert space H_S and its state is represented by a normalized vector ϕ in H_S , that is, $\langle \phi | \phi \rangle = 1$.
- (ii) A measurement Π in S corresponds to an ensemble $\{\Pi_r\}_r$ of projection operators acting on H_S , with $\sum_r \Pi_r = 1$.
- (iii) The Born rule: If we measure Π when system S is in state ϕ , the probability of obtaining result r is given by $Pr(r) = \langle \phi | \Pi_r | \phi \rangle$.
- (iv) The Hilbert space H_{ST} corresponding to the composition of two systems S, T is $H_S \otimes H_T$. The operators used to describe measurements or transformations in system S act trivially on H_T and vice-versa. Similarly, the state representing two independent preparations of the two systems is the tensor product of the two preparations.

All these postulates (including (iv)) can be found in John von Neumann’s “The Mathematical foundations of Quantum mechanics” and in Paul Dirac’s “Principles of Quantum Mechanics”, as well as in modern textbooks on quantum physics (e.g.: Nielsen and Chuang’s). They constitute the definition of quantum theory; any mathematical model violating any of the above postulates cannot be properly regarded as “quantum theory”.

It is true that, for those areas of physics concerned with the observations of a single system, one can do without the last postulate. However, quantum theory, as originally introduced, is a multi-system theory, where postulate (iv) is used to define system composition.

In our paper, we consider an alternative theory, “real quantum theory”, where postulate (i) is replaced by

- (i') To every physical system S , there corresponds a real Hilbert space H_S and its state is represented by a normalized vector ϕ in H_S , that is, $\langle \phi | \phi \rangle = 1$.

This theory is formally equivalent to postulates (i)-(iv), with the restriction that all states and operators have real entries.

There exist infinitely many models that fit the framework of real quantum theory. Some of such models satisfy some super-selection rules, others don't. Our result invalidates them all: no theory satisfying postulates (i'), (ii)-(iv), constrained or not, can explain the statistics observed in the proposed quantum tripartite experiment with two independent sources of entangled states.

In his seminal paper on quantum theory with real numbers, Stueckelberg defines a real mathematical framework to describe single-site quantum experiments. At no point does Stueckelberg consider experiments involving multiple systems, such as a Bell experiment or a demonstration of quantum teleportation. Not surprisingly, a straightforward generalization of Stueckelberg's theory to the multipartite case (the universal real qubit) does not satisfy postulate (iv) *and thus is not quantum theory*. This should come as no surprise, because, in Stueckelberg's times, entanglement and multi-partite experiments were barely discussed. Entanglement, the central piece of quantum physics, was recognized only with the emergence of quantum information science in the 1990's. Our result implies that no generalization of Stueckelberg's theory will, at the same time, satisfy postulates (i'),(ii)-(iv) and lead to the same predictions as standard quantum theory.

Our first manuscript could indeed be misleading. As the referee pointed out, and it is emphasized in the comment arXiv:2103.12740 mentioned in the report, our title and introduction could give the impression that we proved that the empirical content of quantum physics requires complex numbers. As we already explained in the conclusion of our first version, this is obviously not the case. In our rewriting, our first concern was to avoid any misleading statement in this direction. We also added two appendices presenting a more elaborate discussion on this key point.

We now come to a detailed reply to the referee's report.

There are already good theoretical reasons for rejecting real quantum mechanics (e.g. the failure of local tomography) and it is ruled out by several axiomatic approaches to quantum theory e.g. quant-ph/0101012 or Phys. Rev. A 84, 012311.

To our knowledge, there is just one theoretical argument against real quantum theory, namely, local tomography. The quantum axiomatizations presented in quant-ph/0101012 and Phys. Rev. A 84, 012311 (and some others) take this principle as a postulate, so they do not add further insights to that discussion.

The referee shall note, however, that local tomography is not experimentally falsifiable, because any supposed violation can be explained away by arguing that the local measurements conducted on each subsystem were not exhaustive. In addition, not every researcher in quantum foundations agrees that local tomography is such a reasonable principle, as evidenced by the increasing body of work on real quasi-quantum theories.

To make this clear, we now mention this argument in our main text, and discuss local tomography-based arguments in the Supplementary Information.

Although this is a nice result, the paper significantly oversells it. This starts with the title, which taken literally is false, and has already prompted a clarification: arXiv:2103.12740. Furthermore, the abstract says that real quantum physics can reproduce the outcomes of any multipartite experiment, again in contradiction with the title (unless the claim is that the works referred to are wrong). The point is that these previous works use real states that do not respect the usual tensor product structure and are "nonlocal" in a way that ordinary quantum theory is not. This is mentioned in the second to last paragraph of the present paper and this paragraph gives a fair statement of the result (and contradicts other statements in the paper).

Our first version of the paper could indeed be misleading, we are sorry about that. We now changed the title, the abstract and our introduction, in order to make it clear that we do not show that any description of the empirical content of quantum physics requires complex numbers, but rule out real quantum theory, defined through the postulates (i',ii-iv). Making impossible any such confusion was the main objective of our rewriting. This is clarified in the new version of the paper right after we introduce the postulates of real quantum theory, with a more detailed discussion deferred to the SI.

Note that in our new version, the abstract still states that 'real quantum physics' can reproduce the outcomes of any multipartite experiment, as long as the parts share arbitrary real quantum states: the second part of the sentence is necessary to understand the statement. Indeed, real quantum theory can reproduce the statistics of any quantum Bell experiment, where the quantum state shared by the parties is unconstrained. Our result does not contradict this, because the experiment that we propose involves "network scenarios comprising independent quantum state sources and measurements".

Therefore the paper needs significant rewriting to avoid misleading readers. The paper should be rephrased stating that it gives a proposed experimental test that rules out real quantum mechanics (it is much better to talk about what is ruled out since theories can be falsified but not verified). Claims that complex numbers are necessary (without further qualification) should be removed. Without such a rewriting, the results of this work will be misunderstood and I cannot support publication.

We followed the referee's advices by a significant rewriting of our abstract and introduction, and a clear discussion in the SI. The new version of the paper states the postulates of both quantum theory and real quantum theory. We also clarify that the latter cannot reproduce the statistics of certain quantum experiments and hence the quantum framework needs complex numbers to keep the same prediction power.

The authors define CHSH(1,2;1,2) but then use CHSH(1,3;3,4) etc. it would be clearer to define CHSH(x1,x2;y1,y2).

We followed the referee's advice and introduced the symbols x_1 , x_2 , y_1 , y_2 in the definition of CHSH.

Comment on why you have projective measurements

Done.

When you say "any state shared between Alice and Charlie after tracing out Bob...", are you postselecting on Bob's outcomes? I didn't understand exactly what you were referring to. Maybe write down the state to make this clear. Also, what "quantum operations" are you referring to?

We were referring to Alice and Charlie's state before Bob's measurement. We followed the referee's advice and clarified this point.

A link is then made to Fig.1, but the fig has Alice and Bob and not Charlie. Are you imagining a different scenario where Charlie is like Bob in Fig.1 and they just get the state without a third party measuring?

Indeed. Figure 1 is supposed to represent a bipartite Bell experiment, which is what Alice and Charlie are engaged in once Bob conducts his measurement.

Explain the link between (4) and the variants of CHSH_3, i.e., say which variant of CHSH_3 is played depending on b.

Done.

The text is unclear about whether there is a restriction on dimension or not. I think the authors allow the states in real quantum mechanics to have arbitrary (perhaps finite) dimension. If so, this should be stressed. The operators D^A_{zx} etc. are not defined using (3).

There is not restriction on the dimension. We clarified this important point in the new version of the paper.

The "regularized versions of X^A " need to be explained.

This is now explained in the caption of the Figure in the SI.

Reviewer Reports on the First Revision:

Referee #1 (Remarks to the Author):

The authors have made a number of improvements in this paper. Moreover, it is still my opinion that the main results of the paper are sufficiently important and of sufficiently broad interest to merit publication in Nature. My only remaining concern is with the authors' use of the expression "quantum theory." Beginning at the bottom of the first page, the authors take postulates (i) through (iv) to be the defining postulates of quantum theory. I have two objections to this usage.

First, there is a logical problem. If those four postulates really are the defining postulates of quantum theory, then the paper's title, "Quantum theory needs complex numbers," is tautological: quantum theory needs complex numbers merely by definition. The authors themselves say as much on page 2, where they write, "The previous postulates show that, contrary to classical physics, complex numbers (in particular, complex Hilbert spaces) are an essential element of the very definition of quantum theory." I am not making a nit-picking point here. The authors evidently want to use "quantum theory" not to refer to a theory that accords with postulates (i--iv), but rather to refer to any theory that accords EITHER with postulates (i--iv) OR with postulates (i') and (ii--iv). (Certainly this is what "quantum theory" has to mean in the title in order for its claim to be nontautological.) But this use of "quantum theory" cannot be justified by appealing to Refs. [3] and [4], as the authors do at the bottom of page 1. Those references use postulate (i) and not postulate (i')--that is, they use complex numbers from the outset--in defining quantum theory. Thus, for the definition of "quantum theory" that the authors actually need to use in order for the title and much of the rest of the paper to make sense, they offer no sound justification.

My second objection is that the definition of "quantum theory" as a theory necessarily expressed through postulates (i--iv)--even if we leave out the possibility of postulate (i')--is inconsistent with much of the physics literature. To give one example, the phrase "the path-integral formulation of quantum theory" is used quite widely. It is even the title of a chapter in Shankar's textbook, Principles of Quantum Mechanics. Many authors explicitly contrast the path-integral formulation with the Hilbert-space formulation. But if "quantum theory" is defined exclusively to be the Hilbert-space formulation, then "the path-integral formulation of quantum theory" is a self-contradictory phrase. To be sure, complex numbers are central to the path-integral formulation, but my concern here is with the definition of the term "quantum theory." I expect that most authors who formulate quantum theory through postulates (i--iv) would, if asked, readily agree that there are other formulations of quantum theory, and that these alternative formulations also count as quantum theory. The problem is that, in order to exclude from consideration certain formulations of quantum theory that are based on the real numbers--a few such formulations are cited--the authors of the paper under review find themselves needing to define "quantum theory" so that the theory must be expressed in terms of Hilbert space and the tensor product. But this is not how the term is generally understood. We normally understand "quantum theory" to refer to the theory that MAY be expressed by postulates (i--iv) but that can also be expressed in any other equivalent way.

By what expression, then, should the authors refer to the framework defined by postulates (ii--iv) together with the disjunction of postulates (i) and (i')? The authors correctly highlight the crucial role that postulate (iv) plays in their paper. Thus it seems to me that any shorthand expression they choose for the framework they are considering should include an explicit reference to the content of postulate (iv). The expression "the tensor-product formulation of quantum theory" would be a logical choice.

As in my first report, I recommend that the authors choose a title that is not a sentence, since any sentence I can think of that is not misleading would be either too long or too technical to serve as a good title. "The role of complex numbers in quantum theory" would be a fine title.

Again, the authors have done excellent work, but in order to make its significance clear, it is important that they express their results in terms that are consistent with our normal usage.

Referee #3 (Remarks to the Author):

The new version of the main text is significantly improved and makes the statement and significance of the result much clearer. That said, some of the higher-level parts of the text, including the title remain problematic. The understanding of "quantum theory" mentioned by the authors in their reply and in the new version may be well understood by sections of the physics community, but not by all readers and there is still a big danger that the title could be misunderstood. Furthermore, with the understanding mentioned by the authors, it is more or less trivial that quantum theory needs complex numbers because one of the axioms involves a complex Hilbert space and these clearly need complex numbers.

In addition, accepting the understanding of "quantum theory" in the paper, the title would be understood as that the word "complex" is necessary in (i). However, what is actually proven is that replacing "complex" by "real" in (i) can give different predictions. These are not equivalent statements. It is justifiable to say that "complex" cannot be replaced by "real", but it doesn't as far as I can see rule out the possibility of using Hilbert spaces over another field instead. The latter would be required to justify the word "needs" in the title/elsewhere. A more accurate statement of the result is that quantum theory with real Hilbert spaces cannot account for all the predictions of quantum theory, and so can be experimentally falsified.

This is a very interesting statement, and is a non-trivial conclusion which says something important for the foundations of quantum mechanics. I judge that Nature is a bit too broad for it, but that it would be appropriate for publication in the slightly more specialized journal Nature Physics (I think the result is of broad interest in physics, but may not so significant beyond). Ultimately, this is a judgement call and should the editor and other referees disagree, the paper would need some rewriting of the higher-level text before publication.

Typo: on page 5 the word "see" is missing; the text should say "see the Supplementary..."

Author Rebuttals to First Revision:

First of all, we want to thank the referees for their careful reading and the constructive remarks in all the review rounds. We sincerely believe that they have been extremely helpful to improve our manuscript.

Referee #1 (Remarks to the Author):

The authors have made a number of improvements in this paper. Moreover, it is still my opinion that the main results of the paper are sufficiently important and of sufficiently broad interest to merit publication in Nature. My only remaining concern is with the authors' use of the expression "quantum theory." Beginning at the bottom of the first page, the authors take postulates (i) through (iv) to be the defining postulates of quantum theory. I have two objections to this usage.

First, there is a logical problem. If those four postulates really are the defining postulates of quantum theory, then the paper's title, "Quantum theory needs complex numbers," is tautological: quantum theory needs complex numbers merely by definition. The authors themselves say as much on page 2, where they write, "The previous postulates show that, contrary to classical physics, complex numbers (in particular, complex Hilbert spaces) are an essential element of the very definition of quantum theory." I am not making a nit-picking point here. The authors evidently want to use "quantum theory" not to refer to a theory that accords with postulates (i--iv), but rather to refer to any theory that accords EITHER with postulates (i--iv) OR with postulates (i') and (ii--iv). (Certainly this is what "quantum theory" has to mean in the title in order for its claim to be nontautological.) But this use of "quantum theory" cannot be justified by appealing to Refs. [3] and [4], as the authors do at the bottom of page 1. Those references use postulate (i) and not postulate (i')--that is, they use complex numbers from the outset--in defining quantum theory. Thus, for the definition of "quantum theory" that the authors actually need to use in order for the title and much of the rest of the paper to make sense, they offer no sound justification.

The referee is right in pointing out this tautological problem. To correct it, we first define Hilbert space quantum theory in terms of generic Hilbert spaces, without making any reference to whether we use complex or real numbers. The corresponding first postulate (i) associates a Hilbert space to each physical system. This definition is less standard but has appeared before in some works on quantum foundations. Then we define complex and real quantum theory, depending on the field used in the Hilbert spaces, and prove the gap between the two. For that, we introduce the modified postulates (iC) and (iR), which associate a complex or real Hilbert space to each physical system, respectively. This new formulation also clarifies the explanation of our results, as the distinction between complex and real quantum theory is applied systematically through the manuscript, with no further reference to the generic "quantum theory"

My second objection is that the definition of "quantum theory" as a theory necessarily

expressed through postulates (i–iv)--even if we leave out the possibility of postulate (i')--is inconsistent with much of the physics literature. To give one example, the phrase "the path-integral formulation of quantum theory" is used quite widely. It is even the title of a chapter in Shankar's textbook, Principles of Quantum Mechanics. Many authors explicitly contrast the path-integral formulation with the Hilbert-space formulation. But if "quantum theory" is defined exclusively to be the Hilbert-space formulation, then "the path-integral formulation of quantum theory" is a self-contradictory phrase. To be sure, complex numbers are central to the path-integral formulation, but my concern here is with the definition of the term "quantum theory." I expect that most authors who formulate quantum theory through postulates (i--iv) would, if asked, readily agree that there are other formulations of quantum theory, and that these alternative formulations also count as quantum theory. The problem is that, in order to exclude from consideration certain formulations of quantum theory that are based on the real numbers--a few such formulations are cited--the authors of the paper under review find themselves needing to define "quantum theory" so that the theory must be expressed in terms of Hilbert space and the tensor product. But this is not how the term is generally understood. We normally understand "quantum theory" to refer to the theory that MAY be expressed by postulates (i--iv) but that can also be expressed in any other equivalent way.

By what expression, then, should the authors refer to the framework defined by postulates (ii--iv) together with the disjunction of postulates (i) and (i')? The authors correctly highlight the crucial role that postulate (iv) plays in their paper. Thus it seems to me that any shorthand expression they choose for the framework they are considering should include an explicit reference to the content of postulate (iv). The expression "the tensor-product formulation of quantum theory" would be a logical choice.

We also agree with the remark by the referee. To address it, we say before introducing the postulates that they refer to the Hilbert space formulation of quantum theory. Later, after stating our main result, we explicitly mention that it applies to the Hilbert-space formulation of quantum theory, but not to other formulations of the theory, such as path integral, Bohm's theory and other examples listed in the manuscript.

We would also like to remark that what we call "real quantum theory" has actually appeared many times in the quantum foundations literature, often under the very similar name of "real quantum mechanics". The lack of an explicit reference to the validity of the tensor product axiom in the name of the theory has not caused any confusion in the past 30 years, so we do not see the need to complicate our terminology further.

As in my first report, I recommend that the authors choose a title that is not a sentence, since any sentence I can think of that is not misleading would be either too long or too technical to serve as a good title. "The role of complex numbers in quantum theory" would be a fine title.

This will depend on what is decided for the title.

Again, the authors have done excellent work, but in order to make its significance clear, it is important that they express their results in terms that are consistent with our normal usage.

As mentioned, we agree with the remarks by the referee and we have implemented them.

Referee #3 (Remarks to the Author):

The new version of the main text is significantly improved and makes the statement and significance of the result much clearer. That said, some of the higher-level parts of the text, including the title remain problematic. The understanding of "quantum theory" mentioned by the authors in their reply and in the new version may be well understood by sections of the physics community, but not by all readers and there is still a big danger that the title could be misunderstood. Furthermore, with the understanding mentioned by the authors, it is more or less trivial that quantum theory needs complex numbers because one of the axioms involves a complex Hilbert space and these clearly need complex numbers.

To our understanding, this point refers to the tautological point also mentioned by Referee 1 and it has been corrected.

In addition, accepting the understanding of "quantum theory" in the paper, the title would be understood as that the word "complex" is necessary in (i). However, what is actually proven is that replacing "complex" by "real" in (i) can give different predictions. These are not equivalent statements. It is justifiable to say that "complex" cannot be replaced by "real", but it doesn't as far as I can see rule out the possibility of using Hilbert spaces over another field instead. The latter would be required to justify the word "needs" in the title/elsewhere. A more accurate statement of the result is that quantum theory with real Hilbert spaces cannot account for all the predictions of quantum theory, and so can be experimentally falsified.

We agree that there is a slight abuse of terminology and that "complex numbers are needed" is not synonym of "real numbers are not enough". We have added some sentences, such as, e.g., "whether complex numbers are needed within a theory to correctly explain experiments, **or whether real numbers only are sufficient,**" to avoid confusions.

This is a very interesting statement, and is a non-trivial conclusion which says something important for the foundations of quantum mechanics. I judge that Nature is a bit too broad for it, but that it would be appropriate for publication in the slightly more specialized journal Nature Physics (I think the result is of broad interest in physics, but may not so significant beyond). Ultimately, this is a judgement call and should the editor and other referees disagree, the paper would need some rewriting of the higher-level text before publication.

We have indeed taken into account all the remarks by the referee in the new rewriting of the main text.

Typo: on page 5 the word "see" is missing; the text should say "see the Supplementary..."

Implemented.